# A Need to Preserve Ejection Fraction during Heart Failure [note 1]

**DOI:** 10.3390/ijms25168780

**Published:** 2024-08-12

**Authors:** Oluwaseun E. Akinterinwa, Mahavir Singh, Sreevatsa Vemuri, Suresh C. Tyagi

**Affiliations:** 1Department of Physiology, University of Louisville School of Medicine, Louisville, KY 40202, USA; 2Center for Predictive Medicine (CPM) for Biodefense and Emerging Infectious Diseases, University of Louisville, Louisville, KY 40202, USA

**Keywords:** heart failure, reduced ejection fraction, oxidative stress, endothelial dysfunction, mitochondrial dynamics

## Abstract

Heart failure (HF) is a significant global healthcare burden with increasing prevalence and high morbidity and mortality rates. The diagnosis and management of HF are closely tied to ejection fraction (EF), a crucial parameter for evaluating disease severity and determining treatment plans. This paper emphasizes the urgent need to maintain EF during heart failure, highlighting the distinct phenotypes of HF with preserved ejection fraction (HFpEF) and HF with reduced ejection fraction (HFrEF). It discusses the complexities of HFrEF pathophysiology and its negative impact on patient outcomes, stressing the importance of ongoing research and the development of effective therapeutic interventions to slow down the progression from preserved to reduced ejection fraction. Additionally, it explores the potential role of renal denervation in preserving ejection fraction and its implications for HFrEF management. This comprehensive review aims to offer valuable insights into the critical role of EF preservation in enhancing outcomes for patients with heart failure.

## 1. Introduction

Heart failure (HF) is the leading cause of hospitalization worldwide, with over three million new patients diagnosed each year. Being a leading cause of hospitalization in the United States, it is expected that, in the future, nearly six million American adults will have HF, and as a result, the overall incidence of chronic heart ailment is expected to rise to more than eight million [1,2]. HF occurs when the heart cannot pump enough blood to meet the body’s metabolic needs. This results in various symptoms and complications that require hospitalization for treatment. We know that it is the prevalent cause of hospitalization globally, including in the United States [1,2]. The ejection fraction (EF) is a critical parameter for healthcare professionals to evaluate the severity of HF and determine the appropriate course of treatment. The diagnosis and management of HF are intricately linked to the EF, which is paramount in assessing patients with HF [3,4]. The left ventricular ejection fraction (LVEF) is the index of cardiac pumping. It defines the currently accepted heart failure (HF) phenotypes, including HF with reduced ejection fraction (HFrEF), HF with mid-range ejection fraction (HFmrEF), and HF with preserved ejection fraction (HFpEF) [5]. Data from the OPTIMIZE-HF registry show that 49% of patients have HFrEF [6]. HFrEF is a significant public health issue that poses significant morbidity and mortality risks. Despite advancements in treatments over the years, disease morbidity and mortality rates remain high, as evidenced by a five-year survival rate of only 25% following hospitalization for HFrEF [7]. HFrEF is a major contributor to the burden of cardiovascular diseases. Even though HFrEF has been studied extensively, its pathophysiology is still not completely understood due to its complex nature. As the number of patients with heart failure continues to increase over the coming decades, more people are expected to have HFrEF [8,9]. HFpEF and HFrEF are two distinct disorders and do not coexist. EF cannot be both preserved and reduced at the same time and, as such, should be studied and treated separately [10,11]. In clinical presentation, HFrEF is more commonly linked with males, whereas HFpEF is more prevalent in females [12]. Hearts subjected to constant stressors will transition from preserved ejection to reduced ejection fraction due to constant left ventricular remodeling and dilatation, ECM breakdown and restructuring, and inflammatory cell intrusion [3,13,14,15,16]. Patients with heart failure and reduced ejection fraction (HFrEF) have a more severe prognosis compared to those with heart failure and preserved ejection fraction (HFpEF). They have a lower survival rate, a higher mortality, and an increased prevalence of coronary and valve diseases [17,18,19]. These statistics underscore the urgent need for the continued research and development of effective therapeutic interventions to improve the outcomes of patients with this condition and attenuate the transition from preserved to reduced ejection fraction.

## 2. Renal Denervation and Preserving Ejection Fraction

Cardiomyocyte cell death is a hallmark feature observed in HFrEF but not in HFpEF. In patients with HFrEF, damage to cardiomyocytes, as indicated by elevated circulating Troponin-T, leads to a decrease in functioning cardiomyocyte mass combined with excessive fibrotic tissue. The loss of cardiomyocytes in HFrEF occurs through various mechanisms, including apoptosis, necrosis, necroptosis, and autophagy, depending on the underlying cause [2,20,21,22,23,24]. Recent studies have demonstrated that autophagy and necroptosis, a regulated form of necrosis, play a more significant role in cardiomyocyte loss than apoptosis in HFrEF. Notably, even low levels of cardiomyocyte apoptosis have been shown to induce HFrEF in animal models [25,26]. The renal nerves play a crucial role in regulating blood pressure and fluid volume and are closely linked to cardiovascular diseases when not functioning properly. They consist of sympathetic efferent and sensory afferent nerves. Activation of the efferent renal sympathetic nerves triggers renin secretion, sodium absorption, and increased renal vascular resistance, resulting in elevated blood pressure and fluid retention. Renal denervation involves selectively reducing renal sympathetic afferent and efferent signaling through low-dose frequency energy targeted at the renal artery endothelial surface [27,28,29,30,31,32]. When people have multiple health issues like diabetes, central obesity, and hypertension, the sympathetic nervous system (SNS) can become overactive. This overactivation has been associated with higher levels of norepinephrine in patients with HFpEF, indicating its role in the condition. Strong evidence shows SNS upregulation in patients with HFpEF and its harmful effects beyond blood pressure regulation, including cardiac hypertrophy, fibrosis, renal impairment, metabolic dysregulation, and arrhythmogenicity [32,33,34,35,36,37,38]. Excessive SNS activation affects cardiovascular and renal functions, leading to pathological changes such as cardiac hypertrophy, fibrosis, renal dysfunction, metabolic disturbances, and increased risk of arrhythmias, which are commonly found together in patients with HFpEF [32,33,34,35,36,37,38]. A recent study on Zucker Spontaneously Fatty 1 obese rats suggests that early renal denervation can have positive effects on the heart and kidneys in a rodent model of cardiometabolic heart failure with preserved ejection fraction (HFpEF). The research showed that ablating renal sympathetic nerve activity improved cardiac diastolic and vascular function and reduced peri-vascular cardiac fibrosis. Renal denervation also decreased renal injury markers, fibrosis, and renal inflammatory mediators, in part by reducing the synthesis of renal NLR family pyrin domain-containing 3 inflammasome-mediated interleukin 1β [32]. 

It is noteworthy that, in patients with HFrEF, endothelium-mediated vasodilation serves as an independent predictor of cardiac death and hospitalization, signifying the potential protective role of endothelium-derived NO in HFrEF. Endothelial dysfunction is a critical issue that arises due to increased oxidative stress, imbalanced nitric oxide (NO) bioavailability, neurohormonal activation, and vasoconstriction caused by reduced cardiac output [2,39] (Figure 1).

In individuals with HFrEF, cardiac remodeling is primarily driven by damage and a loss of cardiomyocytes, resulting in an imbalanced heart wall structure and eccentric remodeling. This is characterized by left ventricular dilation but with a normal wall thickness. Interestingly, lower levels of insulin-like growth factor 1 (IGF-1), a vasoprotective hormone that stimulates myocardial contraction, have been observed in HFrEF compared to HFpEF. This suggests the reduced protection of cardiomyocytes from hypertrophic and oxidative stress in HFrEF. As a result, distinct types of cardiac hypertrophy are observed in HFrEF and HFpEF due to differences in stimuli and altered cellular signaling [2,20,40,41]. There is evidence to suggest that renal denervation has a positive impact on vascular endothelial function. However, it remains uncertain whether it similarly benefits endocardial endothelial function in individuals with heart failure with a reduced ejection fraction (HFrEF) phenotype [42]. A recent study using a model of chronic volume overload-induced heart failure with arteriovenous fistula (AVF) showed that renal denervation not only preserved ejection transition to reduced ejection fraction but also preserved endocardial endothelial function during heart failure [3]. HFrEF is a complex and multifaceted condition that poses significant morbidity and mortality risks. Despite recent advancements in treatments, the disease remains a significant public health issue, highlighting the need for continued research and development of effective therapeutic interventions to improve outcomes for patients with this condition. Understanding the underlying pathophysiology of HFrEF is crucial for developing novel therapeutic strategies to target the disease’s mechanisms and improve clinical outcomes. Identifying new targets, developing novel therapies, and optimizing existing treatments can reduce the burden of HFrEF and improve the quality of life for patients living with this condition.

## 3. MMP-9 Target and Preserving Ejection Fraction

Matrix metalloproteinases (MMPs) are a group of enzymes that play a crucial role in the breakdown and turnover of the extracellular matrix (ECM). MMPs are vital for maintaining tissue integrity and function and crucial in natural physiological processes such as bone growth and wound healing. However, it is essential to acknowledge that these same MMPs can also be responsible for the onset of severe conditions such as cancer and cardiovascular disease. It is, therefore, imperative to recognize that any disruption in MMP activity can result in a host of disorders that must be urgently addressed [43,44]. The increased activity of MMPs is involved in several disease processes and is known to be involved in the pathogenesis of cardiovascular diseases [45]. Myocardial MMPs are usually found in a dormant state in the heart. However, stressors and oxidative stress activate these resident MMPs in the myocardium (Figure 2).

In the development of chronic heart failure, the activation of matrix metalloproteinase-2 (MMP-2) and matrix metalloproteinase-9 (MMP-9) plays a significant role, particularly in the early and late stages. These enzymes, also known as gelatinase A and gelatinase B, are frequently seen to be overexpressed [46]. MMP-2 is essential in various physiological and pathological processes, including angiogenesis, tissue healing, and inflammation. It is ubiquitously present in all cells and tissues, from heart and endothelial cells to vascular smooth muscle cells, macrophages, and fibroblasts. Thanks to its remarkable and constant activity, MMP-2 is a critical housekeeping gene that efficiently regulates the normal turnover of tissues [47,48]. MMP-9 is essential in managing cardiovascular disease, as it effectively regulates pathological remodeling processes linked to inflammation and fibrosis. Its critical functions involve activating cytokines, breaking down ECM proteins, and stimulating cytokines and chemokines to facilitate tissue remodeling [49]. Studies have shown that strategically removing the *MMP-9 gene* reduces LV dilation and prevents collagen buildup after myocardial infarction in mice, promotes neovascularization, and decreases inflammation during myocardial remodeling after infarction [50,51,52]. Studies have observed that targeting MMP-9 can offer benefits, but it is important to determine its role in the HFpEF phenotype. Is the ablation of the *MMP-9 gene* effective in halting the transition from preserved ejection fraction to reduced ejection fraction during heart failure? Additionally, in cases of HFrEF, there is an increase in mitochondrial regulatory proteins in a model of diabetic cardiomyopathy induced by type 1 diabetes mellitus. This indicates that there may be mitochondrial dysregulation, which could potentially accelerate the death of cardiomyocytes, as observed in HFrEF [53]. Maintaining mitochondrial function and nutrient availability as cellular needs change is crucial, and the process of mitochondria dynamics, which involves fusion and fission mechanisms during mitochondrial movement, plays a vital role in this (Figure 3). Enzymes that hydrolyze guanidine triphosphates (GTPases) regulate these mechanisms, with dynamin-related protein (Drp1) and fission protein (Fis1) primarily involved in mitochondrial fission and mitofusin 1 (Mfn1) and mitofusin 2 (Mfn2) aiding in mitochondria fusion [54,55]. During normal physiological conditions, the fusion and fission processes are well coordinated. However, in conditions like heart failure, these processes can become disrupted [56]. While mitophagy is typically a helpful process for cellular protection, excessive mitochondrial fission can lead to abnormal mitophagy and cell death. This has been identified as a contributing factor in heart failure. Mitochondrial dysfunction is closely associated with mitochondrial fragmentation, often observed during high stress and cellular death. Drp1 activation triggers the fission process, which is responsible for mitophagy [54,55,56]. 

Research has shown that HFrEF is linked to an increase in the Drp1 marker, which indicates mitochondrial fission, and an upregulation of BNIP3 (B-cell lymphoma 2/adenovirus E1B 19 kDa interacting protein 3), a marker for mitochondrial death and mitophagy. HFpEF and HFrEF demonstrate a decrease in mitochondrial area, but HFrEF to a lesser degree than HFpEF. Mitochondrial damage is evident through fragmentation, cristae destruction, and vacuolar degeneration. In HFrEF, the damage to mitochondrial structure and function is particularly significant, indicating a continuous autophagy process [57]. The effect of MMP-9 ablation should be studied in relation to dysregulated mitophagy during heart failure to determine if it helps with the cardiomyocyte death seen in HFrEF and if it helps preserve ejection fraction transition to reduced ejection fraction.

## 4. Conclusions

HFrEF is a significant public health issue that poses significant morbidity and mortality risks. Despite advancements in treatments over the years, disease morbidity and mortality rates remain high. The escalating prevalence of HF presents a pressing and complex challenge that demands immediate attention and rigorous research efforts. With millions of new cases being diagnosed annually, HF imposes a substantial burden on healthcare systems worldwide. It is imperative to underscore the pivotal role of left ventricular ejection fraction (LVEF) assessment in delineating the severity of HF and informing customized treatment strategies. The distinct prognoses associated with HFpEF and HFrEF emphasize the critical need for targeted therapeutic interventions tailored to each HF phenotype. Moreover, it is essential to recognize the intricate nature of HF pathophysiology, particularly the intricate transition from preserved to reduced ejection fraction, which necessitates ongoing and in-depth research endeavors. The long-term prognosis for heart failure with preserved ejection fraction is better than for heart failure with reduced ejection fraction, which has a 42% worse outcome. A study of 6076 discharged patients with heart failure over 15 years found that those with preserved ejection fraction had a higher survival rate compared to those with reduced ejection fraction. Additionally, patients with preserved ejection fraction had lower prevalence rates of coronary artery disease and valve disease compared to those with reduced ejection fraction [18,19]. Specifically, renal denervation’s compelling potential to preserve ejection fraction by modulating the sympathetic nervous system highlights the need for the continued exploration of this approach. Understanding the mechanistic underpinnings of cardiomyocyte loss in HFrEF, the impact of sympathetic nervous system dysregulation in HFpEF, and the involvement of MMP-9 regulation is fundamental to the development of efficacious interventions that can significantly enhance patient outcomes and mitigate the transitions between HF phenotypes.

## Figures and Tables

**Figure 1 ijms-25-08780-f001:**
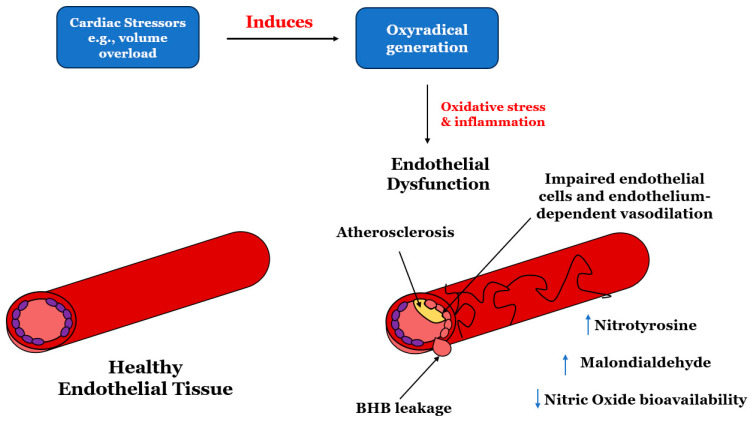
The impact of various cardiac stressors such as volume overload, hypertension, ischemia, etc. on oxyradical generation. For instance, chronic volume overload is a stressor that leads to oxyradical production, resulting in oxidative stress and inflammation. This stress impairs the endothelium’s ability to dilate in response to stimuli due to reduced nitric oxide availability. Endothelial dysfunction, characterized by impaired endothelium-dependent vasodilation, plays a significant role in the development of cardiovascular disorders, including heart failure. Notably, the endothelium’s capacity to dilate blood vessels is a critical predictor of cardiac mortality and hospitalization in individuals with HFrEF.

**Figure 2 ijms-25-08780-f002:**
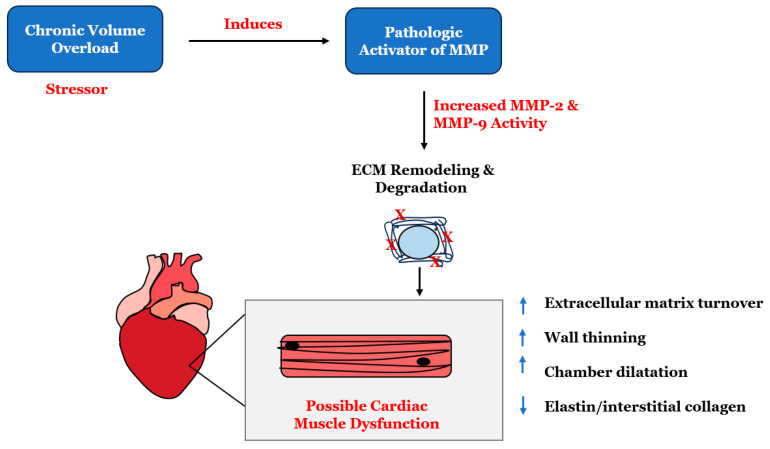
Chronic volume overload, an example of a cardiac stressor and pathologic activator of matrix metalloproteinases (MMPs), leads to the pathological remodeling of the extracellular matrix. In a failing myocardium, latent resident myocardial MMPs are activated, decreasing the elastin-to-interstitial collagen ratio. This, in turn, leads to the formation of an oxidized extracellular matrix. In HFrEF, the loss of fibrillar collagen reduces cardiomyocyte contraction, compromising systolic function.

**Figure 3 ijms-25-08780-f003:**
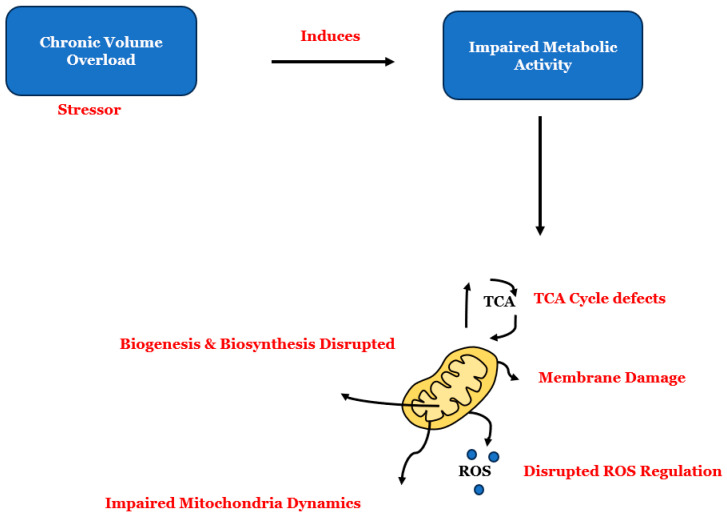
Chronic volume overload is an example of a cardiac stressor and disruptor of mitochondrial function. It affects mitochondrial metabolic activity, including dynamics and bioenergetics. This disruption is closely associated with mitochondrial fragmentation, often observed during high stress and cellular death. In heart failure with reduced ejection fraction (HFrEF), the damage to mitochondrial structure and function is particularly significant, indicating a continuous autophagy process.

## Data Availability

New data were not created or analyzed in this study.

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
