# Peer review of "A Need to Preserve Ejection Fraction during Heart Failureâ€"

_ijms, 2024, doi:10.3390/ijms25168780_

Round 1
Reviewer 1 Report
Comments and Suggestions for Authors
The article's objective is to explore the pathophysiological nexus between chronic volume overload and heart failure with reduced ejection fraction (HFrEF), focusing on oxidative stress, endothelial dysfunction, matrix metalloproteinases activity, and mitochondrial dynamics. However, there are important content issues:
- While the review is somewhat comprehensive, it does not significantly advance the understanding beyond what is already known in the field. The concepts discussed are well-established, with little new hypothesis or groundbreaking theory introduced.
- The underlying pathophysiological systems and pathways are mentioned only superficially without describing or mentioning the molecules involved (except for the MMPs, which are described, albeit without their interacting substrates, e.g., Angiostatin, CD36, Endostatin, Galectin-3, Fibronectin, etc.).
- As a continuation of my previous point, the figures (perhaps with the exception of Figure 1) do not add any novel information, nor do they elucidate molecular pathways in depth. The manuscript would benefit significantly from visual aids that depict the molecular mechanisms, which could help better illustrate the processes and relationships between different pathophysiological factors and molecular pathways.
- The review, while it does discuss some underlying mechanisms, offers limited insight into practical clinical applications or therapeutic interventions that could be derived from this understanding. Thus, the paper could benefit from a more integrated approach that connects molecular findings with clinical outcomes and implications more explicitly.
- Similar to the previous point, although the article mentions potential therapeutic targets and pathways, it lacks a detailed section on how these molecules can be effectively targeted and translated into clinical practice in the future, potentially developing novel therapeutics, which would make the review more impactful for both researchers and clinicians.
Minor Weak Points:
- The transitions between different topics (e.g., from oxidative stress to MMPs) sometimes feel abrupt without a clear linkage or summary that ties the sections into a cohesive narrative.
- The conclusion could be expanded to more effectively state the practical implications of these findings, maybe connecting it to the druggable pathways and mechanisms employed by the pharmacotherapeutic agents already used in HF.
Conclusion:
In conclusion, while the article provides a review of chronic volume overload-induced HFrEF, it largely reiterates known information without offering new insights or hypotheses. The superficial treatment of molecular pathways, limited descriptive figures of molecular mechanisms, abrupt transitions, and an underdeveloped discussion on therapeutic applications diminish its impact.
Author Response
We thank the reviewer for the comments, and per suggestions,. Per advice, we have addressed all the comments one by one (please see below):
Reviewer #1
The article's objective is to explore the pathophysiological nexus between chronic volume overload and heart failure with reduced ejection fraction (HFrEF), focusing on oxidative stress, endothelial dysfunction, matrix metalloproteinases activity, and mitochondrial dynamics. However, there are important content issues: -
Comment # 1: While the review is somewhat comprehensive, it does not significantly advance the understanding beyond what is already known in the field. The concepts discussed are well-established, with little new hypothesis or groundbreaking theory introduced.
Reply: Thank you for your comment. Our review manuscript focuses on examining how ejection fraction is preserved in heart failure patients prior to transitioning to reduced ejection fraction, using existing knowledge. Our goal is to consolidate scattered information from various sources into a single paper, rather than proposing new hypotheses or theories. This revision is evident in our updated title and throughout the manuscript, which has been extensively edited, revised, and expanded to better address this issue for our readers.
Comment # 2: The underlying pathophysiological systems and pathways are mentioned only superficially without describing or mentioning the molecules involved (except for the MMPs, which are described, albeit without their interacting substrates, e.g., Angiostatin, CD36, Endostatin, Galectin-3, Fibronectin, etc.).
Reply: We appreciate the reviewer's feedback. Following the advice, we have revised our manuscript and incorporated new sections to better match the adjusted title. Throughout the text, we have included molecule names as necessary, ensuring they support rather than distract from the manuscript's primary focus.
Comment # 3: As a continuation of my previous point, the figures (perhaps with the exception of Figure 1) do not add any novel information, nor do they elucidate molecular pathways in depth. The manuscript would benefit significantly from visual aids that depict the molecular mechanisms, which could help better illustrate the processes and relationships between different pathophysiological factors and molecular pathways.
Reply: We appreciate your comment. As per the updated title of our manuscript, our primary aim is to analyze the preservation of ejection fraction in heart failure patients before it transitions to reduced ejection fraction, using existing literature and knowledge. Additionally, we have thoroughly revised and edited the manuscript, incorporating new sections to better correspond with the revised title.
Comment # 4: The review, while it does discuss some underlying mechanisms, offers limited insight into practical clinical applications or therapeutic interventions that could be derived from this understanding. Thus, the paper could benefit from a more integrated approach that connects molecular findings with clinical outcomes and implications more explicitly.
Reply: In our revised manuscript, we have introduced new sections that are closely aligned with the updated title, focusing on the preservation of ejection fraction in heart failure. We appreciate the reviewer's feedback on this matter.
Comment # 5: Similar to the previous point, although the article mentions potential therapeutic targets and pathways, it lacks a detailed section on how these molecules can be effectively targeted and translated into clinical practice in the future, potentially developing novel therapeutics, which would make the review more impactful for both researchers and clinicians.
Reply: Once again, we appreciate the reviewer's comment. Our revised manuscript now includes additional sections that better reflect the modified title.
Minor Weak Points:
Commet #1: The transitions between different topics (e.g., from oxidative stress to MMPs) sometimes feel abrupt without a clear linkage or summary that ties the sections into a cohesive narrative.
Reply: We appreciate the reviewer's feedback, and our revised version now emphasizes preserving the ejection fraction to align more closely with the title.
Commet #2: The conclusion could be expanded to more effectively state the practical implications of these findings, maybe connecting it to the druggable pathways and mechanisms employed by the pharmacotherapeutic agents already used in HF.
Reply: Again, we appreciate the reviewer's comment. As mentioned earlier, the manuscript has been revised to align with the updated title.
Reviewer 2 Report
Comments and Suggestions for Authors
I read with difficult the paper from Akinterinwa and coworkers.
The manuscript titled “Dysfunction, and Mitochondrial Dynamics: Unveiling the Nexus” is based on some misleading concepts.
Introduction section line 38
the left ventricular ejection fraction is “ a measure of strength of the heart”, the LVEF is a cardiac pump index subject to the influence of preload and afterload, therefore it cannot provide measure the myocardium contractility that is the index of the cardiac strength.
Line 43
“The left ventricular EF is the primary parameter used for diagnosing….”
The sentence should be changed as follow “the LVFEF is the cardiac pump index, on the basis of predefined cutoff value, defines the current accepted HF phenotypes, HFrEF, HFmrEF and HFpEF.(Palazzuoli A J Clin Med. 2023 Jan 15;12(2):693. doi: 10.3390/jcm12020693)
Line 62 -63
“A decreased cardiac contractility is a hallmark of myocardial failure”.
This assertion is untrue. In the PARAGON HF study the female gender that had a LVEF > 60% at the enrollment enjoyed superior benefit from ARNI administration (McMurray JJV https://doi.org/10.1161/CIRCULATIONAHA.119.044491)
“This decreased contractility results in an increased end-systolic volume and increased left ventricular diastolic filling pressure”. The Authors should explain the increased diastolic volume in presence of impaired systolic function is the primary mechanism that allows the cardiac function to engage the preload reserve in matching the afterload increase.
Line 73
“Congestive heart failure (CHF) results when a HF patient develops intravascular volume overload, and the heart is incapable of handling the extra volume. Thus, edema develops in tissues like the lung etc..!)
The pathophysiology of congestive heart failure is far more complex than the sentence addresses. The Authors ignore the pivotal role of baroreceptor in governing the cardiovascular balance and the effect on renal sympathetic response.
Line 79
-“A heart subjected to chronic volume overload will transition to HFrEF due to constant left ventricular remodeling and dilatation, ECM breakdown and restructuring, and inflammatory cell intrusion [3, 20-23].
The Authors address “volume overload” without to explain what they mean by adopting such condition as a unique (??) HF cause.
What do they mean speaking of volume overload”? mitral regurgitation, aortic valve regurgitation? Interventricular septal communication or what?? Just the arteriovenous fistula??, and what about ischemic heart disease and systolic dysfunction or aortic stenosis with Low flow Low gradient? by removing the cause of systolic impairment (ischemia and left ventricular outflow obstruction) the patient outcome improves in both conditions.
Line 95
- As the number of heart failure patients continue to increase over the coming decades, more people are expected to have
HFrEF.
The sentence should also address the today HFpEF patient population is larger in comparison to the one of HFrEF patients and in the incoming future the HFpEF grow will be higher.
The pathophysiology of HFpEF and HFrEF is different despite the two conditions share nuances. On note progression from HFpEF to HFrEF entails worse prognosis, while reversion from lower LVEF to higher LVEF mirrors outcome improvement. The point deserves appropriate consideration in the manuscript
- SECTION 3 , line 144 and following
- “Arteriovenous fistula (AVF) is one of the experimental models of volume-overload CHF and has been used in studying the pathophysiology of heart failure”.
The arteriovenous fistula is just a model of HF based on diastolic overload. It cannot represent the multiple causes of heart failure development and evolution as many other mechanisms are involved in etiology and progression of cardiac pump failure. One simple example is provided by all conditions causing left ventricular outflow obstruction.
To address the complex interaction between glucose metabolism, HF ensue and metabolic stressors it’s worthy to recall that acute decompensation is prevented by abolishing the insulin myocardial receptors in a HF animal model of aortic outflow obstruction (Simons DJ Am J Physiol Heart Circ Physiol 300: H374–H381, 2011; doi:10.1152/ajpheart.01200.2009)
In this section the issue of neurohormonal response is tackled with inappropriate simplification. The topic deserves to be thoroughly explored as the related literature is extensive and provides pillar knowledge that cannot be underestimated. The reactive oxygen species phenomenon described in congestive (overload) HF condition must be analyzed in the context of the neurohormonal activation led by the HF hemodynamic derangement generated by cardiac function impairment and discussed on reference to associated conditions that can amplify the oxygen species production such as diabetes.
On top of this the reactive oxygen species production occurs in different models of systolic impairment leading to HF development.
(Hartupee, J., Mann, D. Neurohormonal activation in heart failure with reduced ejection fraction. Nat Rev Cardiol 14, 30–38 (2017). https://doi.org/10.1038/nrcardio.2016.163; Bloom, M., Greenberg, B., Jaarsma, T. et al. Heart failure with reduced ejection fraction.Nat Rev Dis Primers 3, 17058 (2017). https://doi.org/10.1038/nrdp.2017.58).
Author Response
We thank the reviewer for the comments, and per suggestions, we have addressed all the comments one by one, as listed below:
Reviewer #2
Comment #1: Introduction section line 38
the left ventricular ejection fraction is “ a measure of strength of the heart”, the LVEF is a cardiac pump index subject to the influence of preload and afterload, therefore it cannot provide measure the myocardium contractility that is the index of the cardiac strength.
Reply: Thank you for the comment. As advised, we have made changes and corrected the text (line 37-40).
Comment #2: Line 43
“The left ventricular EF is the primary parameter used for diagnosing….”
The sentence should be changed as follow “the LVFEF is the cardiac pump index, on the basis of predefined cutoff value, defines the current accepted HF phenotypes, HFrEF, HFmrEF and HFpEF. (Palazzuoli A J Clin Med. 2023 Jan 15;12(2):693. doi: 10.3390/jcm12020693)
Reply: Per advice, we have made changes and corrected the text.
Comment #3: Line 62 -63
“A decreased cardiac contractility is a hallmark of myocardial failure”.
This assertion is untrue. In the PARAGON HF study the female gender that had a LVEF > 60% at the enrollment enjoyed superior benefit from ARNI administration (McMurray JJV https://doi.org/10.1161/CIRCULATIONAHA.119.044491)
Reply: We have made changes per advice. Thank you for the comment.
Comment #4: “This decreased contractility results in an increased end-systolic volume and increased left ventricular diastolic filling pressure”. The Authors should explain the increased diastolic volume in presence of impaired systolic function is the primary mechanism that allows the cardiac function to engage the preload reserve in matching the afterload increase.
Reply: Thank you for the suggestion. In the revised version, we have emphasized the preservation of ejection fraction. Consequently, we have rewritten and added new sections to the paper to better align with the new title.
Comment #4: Line 73
“Congestive heart failure (CHF) results when a HF patient develops intravascular volume overload, and the heart is incapable of handling the extra volume. Thus, edema develops in tissues like the lung etc.!)
The pathophysiology of congestive heart failure is far more complex than the sentence addresses. The Authors ignore the pivotal role of baroreceptor in governing the cardiovascular balance and the effect on renal sympathetic response.
Reply: We appreciate the reviewer's comment. While congestive heart failure is indeed complex, delving into details about baroreceptors and the renin-angiotensin-aldosterone system (RAAS) would diverge from the current theme of our manuscript. In our previous research on an AVF-induced heart failure model, we have observed dysfunction in neurohormonal responses, as documented in our published work and the literature available. Despite volume overload, this system remains active, leading to continuous compensatory activities such as remodeling, dilation, and immune cell infiltration in cardiac tissue. Ultimately, this process progresses from preserved to reduced ejection fraction. In response to the feedback, we have focused our revised manuscript on preserving ejection fraction. We have accordingly revised the text and introduced new sections to better align with the updated title. Thank you for your input.
Comment #5: Line 79
-“A heart subjected to chronic volume overload will transition to HFrEF due to constant left ventricular remodeling and dilatation, ECM breakdown and restructuring, and inflammatory cell intrusion [3, 20-23].
The Authors address “volume overload” without to explain what they mean by adopting such condition as a unique (??) HF cause.
What do they mean speaking of volume overload”? mitral regurgitation, aortic valve regurgitation? Interventricular septal communication or what?? Just the arteriovenous fistula??, and what about ischemic heart disease and systolic dysfunction or aortic stenosis with Low flow Low gradient? by removing the cause of systolic impairment (ischemia and left ventricular outflow obstruction) the patient outcome improves in both conditions.
Reply: Thank you for your comment. You are correct in noting that there are various factors contributing to heart failure. However, for this review manuscript, we have chosen to concentrate specifically on volume overload as a model for the transition from preserved to reduced ejection fraction. We acknowledge the significance of other factors such as mitral regurgitation, aortic valve regurgitation, interventricular septal communication, arteriovenous fistula, ischemic heart disease, and other conditions leading to systolic dysfunction. Nevertheless, to align better with the new title, we have adjusted the manuscript by removing sections and narrowing our focus to collect and review literature on preserved ejection fraction.
Comment #6: Line 95
- As the number of heart failure patients continue to increase over the coming decades, more people are expected to have HFrEF. The sentence should also address the today HFpEF patient population is larger in comparison to the one of HFrEF patients and in the incoming future the HFpEF grow will be higher.
The pathophysiology of HFpEF and HFrEF is different despite the two conditions share nuances. On note progression from HFpEF to HFrEF entails worse prognosis, while reversion from lower LVEF to higher LVEF mirrors outcome improvement. The point deserves appropriate consideration in the manuscript.
Reply: We appreciate the comment and per advice we have modified the text (Line 37-48). Thank you.
Comment #7: - SECTION 3 , line 144 and following
- “Arteriovenous fistula (AVF) is one of the experimental models of volume-overload CHF and has been used in studying the pathophysiology of heart failure”. The arteriovenous fistula is just a model of HF based on diastolic overload. It cannot represent the multiple causes of heart failure development and evolution as many other mechanisms are involved in etiology and progression of cardiac pump failure. One simple example is provided by all conditions causing left ventricular outflow obstruction.
Reply: We appreciate the feedback and would like to clarify that we did not claim to cover all the causes of heart failure. Our focus remains on arteriovenous fistula (AVF) as it is one of the experimental animal models that has demonstrated a transition from preserved to reduced ejection fraction. As mentioned earlier, we have revised and removed sections from the manuscript to better align with the new title.
Comment #8: To address the complex interaction between glucose metabolism, HF ensue and metabolic stressors it’s worthy to recall that acute decompensation is prevented by abolishing the insulin myocardial receptors in a HF animal model of aortic outflow obstruction (Simons DJ Am J Physiol Heart Circ Physiol 300: H374–H381, 2011; doi:10.1152/ajpheart.01200.2009).
Reply: Appreciate the suggestion, and now we have rewritten the section(s) of the manuscript to align better with the new title.
Comment #9: In this section the issue of neurohormonal response is tackled with inappropriate simplification. The topic deserves to be thoroughly explored as the related literature is extensive and provides pillar knowledge that cannot be underestimated. The reactive oxygen species phenomenon described in congestive (overload) HF condition must be analyzed in the context of the neurohormonal activation led by the HF hemodynamic derangement generated by cardiac function impairment and discussed on reference to associated conditions that can amplify the oxygen species production such as diabetes.
Reply: Thank you for the advice. In the revised version of the manuscript, we have rewritten and removed some sections to better align with the new title.
Comment #10: On top of this the reactive oxygen species production occurs in different models of systolic impairment leading to HF development. (Hartupee, J., Mann, D. Neurohormonal activation in heart failure with reduced ejection fraction. Nat Rev Cardiol 14, 30–38 (2017). https://doi.org/10.1038/nrcardio.2016.163; Bloom, M., Greenberg, B., Jaarsma, T. et al. Heart failure with reduced ejection fraction. Nat Rev Dis Primers 3, 17058 (2017). https://doi.org/10.1038/nrdp.2017.58).
Reply: We agree with the reviewer. In our thoroughly revised manuscript, we have focused on arteriovenous fistula (AVF) as a model of ejection fraction (EF) transition. Consequently, we have rewritten and removed sections of the manuscript to better align with our new title.
Reviewer 3 Report
Comments and Suggestions for Authors
Reviewer report 
 
 
Chronic Volume Overload and Heart Failure with Reduced Ejection Fraction- Insights into Oxidative Stress, Endothelial Dysfunction, and Mitochondrial Dynamics: Unveiling the Nexus
This is a well-written review article showing the relationship between the pathophysiological condition of heart failure in the context of HFpEF and HFrEF. Later the article focuses lights on various molecular-level pathological events of heart failure like oxidative stress, endothelial dysfunction, and the role of matrix metalloproteases in heart failure. The article describes the causative factors in various sections along with the associated figures.
Please explain the entire review in the form of a graphical abstract with a summary and conclusion.
How does this review differ from the reported data in the literature any novel things reported, or any mechanisms reported please list out.
How HFrEF is related to left ventricular posterior wall thickness for systole and diastole as well as left ventricular anterior wall thickness for systole and diastole please explain briefly.
Please list out the studies related to Heart Failure with Reduced Ejection Fraction in the form of a table in context to pre-clinical and clinical settings.
Does HFpEF and HFrEF have gender-based differences? does the pathophysiological outcome differ in males and females?

Author Response
We thank the reviewer for the comments, and per suggestions, we have addressed all the comments one by one, as written below:
Reviewer #3
Chronic Volume Overload and Heart Failure with Reduced Ejection Fraction- Insights into Oxidative Stress, Endothelial Dysfunction, and Mitochondrial Dynamics: Unveiling the Nexus.
General comment: This is a well-written review article showing the relationship between the pathophysiological condition of heart failure in the context of HFpEF and HFrEF. Later the article focuses lights on various molecular-level pathological events of heart failure like oxidative stress, endothelial dysfunction, and the role of matrix metalloproteases in heart failure. The article describes the causative factors in various sections along with the associated figures.
Reply: We are delighted to hear that the reviewer appreciated our manuscript, and we sincerely value their encouragement. Thank you very much!
Comment # 1: -Please explain the entire review in the form of a graphical abstract with a summary and conclusion.
Reply: Thank you for your comment. We have thoroughly revised our manuscript, removed some sections and added new ones to better align with the modified title.
Comment # 2: -How does this review differ from the reported data in the literature any novel things reported, or any mechanisms reported please list out.
Reply: We appreciate the reviewer's comments and would like to clarify that our manuscript does not focus on presenting groundbreaking hypotheses or theories. Instead, it aims to compile and examine the preservation of ejection fraction in heart failure patients before the transition to reduced ejection fraction, analyzing existing knowledge comprehensively. Our approach addresses the issue from the perspectives of oxidative stress, endothelial dysfunction, and mitochondrial dysfunction. These crucial aspects are thoroughly discussed in the manuscript, aligning perfectly with the modified title
Comment # 3: -How HFrEF is related to left ventricular posterior wall thickness for systole and diastole as well as left ventricular anterior wall thickness for systole and diastole please explain briefly.
Reply: Thank you for the comment. We have revised the manuscript to better align with the new title.
Comment # 4: Please list out the studies related to Heart Failure with Reduced Ejection Fraction in the form of a table in context to pre-clinical and clinical settings.
Reply: Thank you for the comment. As mentioned above, we have now thoroughly revised our manuscript and have also added new sections to better align with the modified title.
Comment # 5: -Does HFpEF and HFrEF have gender-based differences? does the pathophysiological outcome differ in males and females?
Reply: We appreciate the reviewer’s comment. Indeed, heart failure with preserved ejection fraction (HFpEF) and heart failure with reduced ejection fraction (HFrEF) exhibit gender-based differences. Research suggests that HFpEF is more prevalent in women, whereas HFrEF appears to be more common in men. Furthermore, the clinical presentation, underlying pathophysiology, and response to treatment may vary between genders in both types of heart failure. Further studies are needed to enhance our understanding and address these gender-based distinctions in the management of heart failure.